# Pharmacotherapy Based on ACE2 Targeting and COVID-19 Infection

**DOI:** 10.3390/ijms23126644

**Published:** 2022-06-14

**Authors:** Antonio Vitiello, Francesco Ferrara

**Affiliations:** 1Pharmaceutical Department, Usl Umbria 1, Via XIV Settembre, 06132 Perugia, Italy; antonio.vitiello2@uslumbria1.it; 2Pharmaceutical Department, Asl Napoli 3 Sud, Dell’amicizia Street 22, 80035 Naples, Italy

**Keywords:** ACE2, COVID-19, SARS-CoV-2, RAS

## Abstract

The new SARS-CoV-2 coronavirus is responsible for the COVID-19 pandemic. A massive vaccination campaign, which is still ongoing, has averted most serious consequences worldwide; however, lines of research are continuing to identify the best drug therapies to treat COVID-19 infection. SARS-CoV-2 penetrates the cells of the host organism through ACE2. The ACE2 protein plays a key role in the renin–angiotensin system (RAS) and undergoes changes in expression during different stages of COVID-19 infection. It appears that an unregulated RAS is responsible for the severe lung damage that occurs in some cases of COVID-19. Pharmacologically modifying the expression of ACE2 could be an interesting line of research to follow in order to avoid the severe complications of COVID-19.

## 1. COVID-19

The COVID-19 pandemic has so far caused about 520,912,257 infections and 6,272,408 deaths [1,2,3]. The massive vaccination campaign that is still taking place worldwide, thanks mainly to mRNA vaccines, has prevented the number of deaths and severe cases from being higher; however, research efforts continue to identify the best drug treatments and cures for treating the COVID-19 viral infection. In most cases, the infection has an asymptomatic or mildly symptomatic course. The most common clinical manifestations of COVID-19 infection include loss of sense of smell, fatigue, fever, cough, and dyspnea [4,5,6]. In severe cases, injury to the lungs and heart can occur, caused by an abnormal and dysregulated response of the inflammatory/immune system induced by a massive and sudden release of proinflammatory mediators such as cytokines and chemokines [7,8,9,10]. In the early stages of the pandemic, several pharmacological treatments were proposed for COVID-19 infection, including anti-inflammatory/immunomodulatory agents, anticoagulants, convalescent plasma, and antivirals indicated for other diseases [11,12,13,14]. Today, there are several antivirals directed against SARS-CoV-2. The SARS-CoV-2 coronavirus penetrates cells through the ACE2 glycoprotein, which is expressed in different tissues of the body [15,16,17]. In this sense, the homeostasis of the renin–angiotensin system is another risk factor underlying the pathogenesis of COVID-19, since angiotensin-converting enzyme 2 (ACE2) is the predominant receptor through which the SARS-CoV-2 virus enters cells and infects them.

This probably also shows that COVID-19 is not only a respiratory disease but also a multisystem disease that can cause neurological, cardiovascular, and reproductive system damage [18,19,20]. An altered ACE/ACE2 expression ratio could contribute to severe outcomes in patients with COVID-19 [21], as is the case with cardiovascular disease. ACE2 plays a key role in the RAS renin-angiotensin system.

## 2. The Role of ACE2

The renin-angiotensin system (RAS) maintains blood pressure and electrolyte balance in the body and has also been implicated in the pathogenesis of acute respiratory distress syndrome (ARDS). The RAS operates through two axes: the classical angiotensin-converting enzyme (ACE)/Angiotensin (Ang) II/Ang II type 1 receptor (AT1) axis and the non-classical ACE2/Ang 1–7/Mas receptor (MasR) axis. These two pathways have opposite functions: while the former is associated with impaired respiratory status, the latter plays a protective role in ARDS. SARS-CoV-2 virus uses ACE2 as its receptor-binding domain for endocellular penetration [22]. ACE2 is a carboxypeptidase and a type I transmembrane protein with an N-terminal extracellular domain containing the active site. ACE2 is a key regulator of the renin–angiotensin system (RAS). ACE2 receptors are expressed in different tissues of the body, particularly in the lungs, heart, intestine, and testes [23]. In the lungs, the highest concentration of ACE2 has been identified in type II pneumocytes [24]. This may suggest why the lung is the most vulnerable target organ for COVID-19 infection. ACE2 is also expressed as a coreceptor in intestinal epithelial cells, where it mediates the role of nutrient absorption [25]. ACE2 has been found to be expressed on cell membranes and in circulation in soluble form. The role that ACE2 plays in the various stages of COVID-19 infection is still not entirely clear. Some evidence shows a variation in the modulation of RAS and ACE2 during the various stages of infection. After ACE2 binding and SARS-CoV-2 entry into target cells, shedding of host ACE2 receptors occurs, which may alter RAS tissue homeostasis, with important implications for the severity of COVID-19. ACE2 appears to play a protective role on the lungs during viral infection. This evidence suggests the use of ACE2-targeted therapeutic approaches. ACE2 mediates the conversion of Ang-II to Ang 1–7 and Ang-I to Ang 1–9 (Figure 1). The biological effects of Ang-II mediated by AT1 receptors are vasoconstrictive, hypertrophic, and proinflammatory, increasing oxidative stress and coagulation, biological effects that, if dysregulated and uncontrolled, can complicate the course of COVID-19 infection. In addition, Ang-II can induce increased inflammation through the production of IL-6, tumor necrosis factor (TNF)-α, and other inflammatory cytokines [26,27,28]. ACE2 causes the degradation of Ang-II and the formation of Ang 1–7. Ang 1–7 has opposite biological effects to Ang-II through MasR and AT2-r, such as anti-inflammatory and antifibrosis, antiplatelet, and antihypertrophic effects. In support of this fact, recent evidence has shown that the serum of highly exposed but uninfected individuals has the ability to neutralize SARS-CoV-2, probably mediated by soluble ACE2, and that lower levels of ACE2, are present, although only marginally significantly, in more severe patients [29]. Further evidence shows that sACE2 levels were lower in severe and moderate COVID-19 patients than in mild subjects at the time of admission. Within 5–7 days, as patients recovered, sACE2 levels increased in all moderate patients, but only in half of the patients with severe disease [30]. The description of these molecular and biological aspects suggests the use of therapeutic solutions based on ACE2 targets [31,32].

## 3. Pharmacological Approach on ACE2

SARS-CoV-2 infection impairs endothelial function through the downregulation of ACE2, leading to lung injury. In addition, among other pathophysiological aspects of COVID-19, stress on the RAS also plays a role in the development and severity of the disease. However, it is unclear at what stage of COVID-19 disease ACE2 is downregulated. Increased expression of ACE2 could prove to be of therapeutic benefit in COVID-19 infection. Commonly used RAS-modulating pharmacological agents such as ACEi increase ACE2 levels, whereas the use of ARBs causes an increase in the expression and activity levels of ACE and ACE2 [33,34]. Increased ACE2 might be useful in the late stages of infection to counteract the hyperinflammatory and hyperphybrotic state of lung tissue. Finally, increased ACE2 could lead to increased bradykinin degradation, further preventing pro-inflammatory, pro-oxidant, and profibrotic effects [35]. Treatment with an rhACE2-soluble form of ACE2 could prove useful as a decoy effect for SARS-CoV2 and decrease cellular entry of the virus, thereby hindering viral infection [36]. The administration of recombinant soluble human ACE2(rhACE2) has shown good efficacy in subjects with acute respiratory distress syndrome (ARDS) [37]. From a molecular pharmacological point of view, the administration of rhACE2 activates the Ang 1–7 and Ang 1–9 synthesis pathway of the RAS system (nonclassical pathway) by decreasing Ang II levels with a tendency to lower proinflammatory cytokine concentrations. Evidence shows that [38] the administration of rhACE2 blocks the early stages of SARS-CoV-2 infection by preventing SARS-CoV-2 from entering cells and increases the efficacy of remdesivir activity when administered in combination [39,40]. The rhACE2 can act as a “decoy”. Their molecular “decoy effect” binds the spike protein SARS-CoV-2 so tightly that it competes with true human ACE2 and prevents the virus from carrying on the infectious process (Figure 2).

Not only does the sACE2 function as a decoy receptor, its catalytic activity as an enzyme in the renin-angiotensin-aldosterone (RAS) system could be highly beneficial to decrease levels of Ang II. Engineered variants of sACE2 may possess catalytic activity and contribute to the conversion of Ang II to Ang 1,7 to treat respiratory distress as well [41]. Another interesting line of research is the administration of Ang 1–7 peptide; clinical trials in this direction are ongoing. The sites of interaction between ACE2 and SARS-CoV-2 represent important pharmacological targets to synthesize compounds directed against this site of action. In addition, vitamin D administration raises ACE2 levels [42], potentially representing a role in combating severe complications from COVID-19. In addition, viral delivery systems using adenoviruses, adeno-associated viruses, or lentiviruses have been used as approaches to increase ACE2 expression in vivo in the central nervous system and a variety of peripheral tissues. Several compounds named “ACE2 activators” are being investigated to amplify ACE2 [43]. Finally, an interesting pharmacological approach could be allosteric modulation. It is hypothesized that SARS-CoV-2 may lose the ability to infect new host cells due to an allosterically impaired interaction between the ACE2 receptor and the viral RBD. Drugs that act by allosteric modulation, such as allosteric enzymes, bind to sites other than the active site and often alter the shape of the active site itself and its affinity [44]. Finding pharmacological compounds that use allosteric modulation to alter the ACE2 binding site with RBD SARS CoV-2 could be an interesting line of research. In other words, due to altered biophysical properties of the ACE2 receptor, viral RBD may lose or enhance the degree of affinity toward the ACE2 receptor [45]. Similarly, the alteration of the biophysical properties of the ACE2 receptor could be possible following allosteric drug binding, which could disrupt the interactions between ACE2 and the RBD of SARS-CoV-2. Some evidence has shown that the binding of a drug to the allosteric site of the ACE2 receptor can decrease biophysical interactions of the ACE2 and viral RBD. 

Specifically for some drugs such as dexamethasone (DEX), chloroquine (CQ), and telmisartan (TLS), binding to an allosteric site with a conformational shift of ACE2 can disrupt interactions between the SARS-CoV-2 spike protein and human ACE2 [45]. However, the binding of a drug to an allosteric site of ACE2 can also reduce the conversion of angiotensin-I and -II enzyme substrate. Therefore, altering the biophysical properties of the ACE2 receptor by modulating an allosteric site of ACE2 could be a promising strategy against COVID-19. This aspect will need to be explored in detailed in future research. Finally, current evidence has not provided evidence for a significant association between ACEI/ARB treatment and COVID-19 mortality. ACEIs/ARBs should not be withdrawn in the absence of formal contraindications [46].

## 4. Conclusions

A pharmacological approach based on increasing ACE2 may reduce the detrimental actions mediated by the stimulation of AT1r by Ang2 and increase the benefits of the stimulation of MASr activated by Ang 1–7 and Ang 1–9. These benefits are potentially useful in combating COVID-19 infection and reducing the risk of serious complications, such as hyperinflammatory state and tissue hyperfibrosis, and helping to mitigate pulmonary, cardiac, and renal damage caused by COVID-19. Clinical evidence is needed to generate useful data to confirm this interesting line of research.

## Figures and Tables

**Figure 1 ijms-23-06644-f001:**
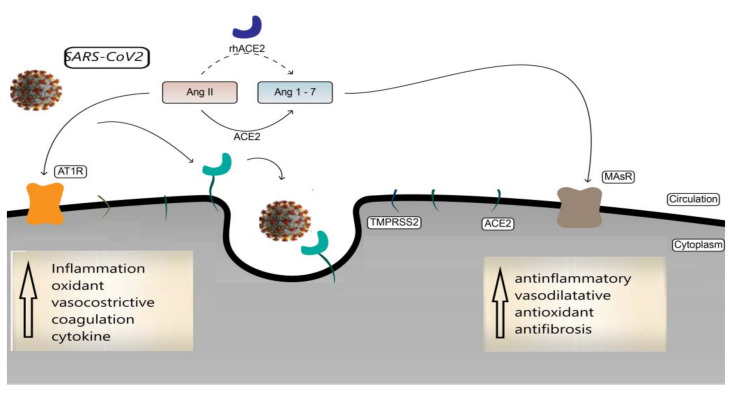
Schematic representation of the correlation between SARS-CoV-2 and ACE2: SARS-CoV-2 penetrates cells through binding of the viral spike protein (spike; S) to angiotensin-converting enzyme 2 (ACE2). ACE2 converts angiotensin (Ang)-II into Ang 1–7, which has biological effects mediated by MASr activation (antifibrotic, antioxidant, and antihypertrophic) as opposed to Ang-II mediated by AT1 receptors (prooxidant, hypertrophic, vasoconstrictive, and hyperinflammatory).

**Figure 2 ijms-23-06644-f002:**
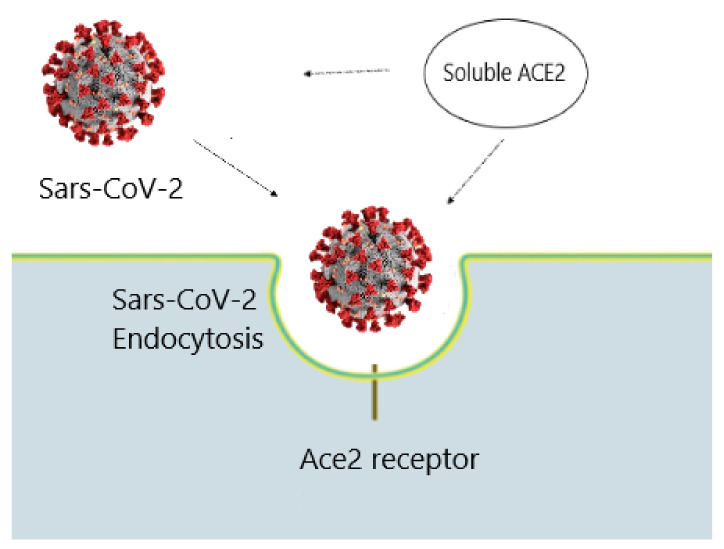
Soluble ACE2 binding prevents SARS-CoV-2 attachment to membrane ACE2, reducing viral endocellular penetration.

## Data Availability

Full availability of data and materials.

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
