# Peer review of "Pharmacotherapy Based on ACE2 Targeting and COVID-19 Infection"

_ijms, 2022, doi:10.3390/ijms23126644_

Round 1

Reviewer 1 Report

Major concerns

  • 17 out of 37 references of this manuscript are selfcitations. Many of them are not research articles but only point of view and letter to the editor. Authors must discuss original articles.  
  • What is the aim of this study? Looking at the title, it seems that authors suggest possible therapies to target ACE2 in COVID-19 but this hypothesis is lost in the manuscript since they do not explain how these treatment can have a beneficial effect in COVID-19 patients. Moreover, authors conclude that "a pharmacological approach based on increasing ACE2 may reduce the detrimental actions mediated by stimulation of AT1r by Ang2, and increase the benefits of stimulation of MASr activated by Ang 1-7 and Ang 1-9". Authors must explain how increased ACE2 (it is not clear which form of ACE2 they mean) can induce this beneficial effects avoinding the side effects of these treatments. Paradoxically, use of ACEis and ARBs (commonly used in non-COVID-19 patients to treat hypertension) can increase ACE2 expression making these patients more susceptible to SARS-CoV-2 infection since the virus use ACE2 as receptor to enter into the cells.

Minor concerns

  • "2. The role of ACE-2": due to the fact that ACE2 is also expressed in many tissues, authors need to specify that SARS-CoV-2 can also cause non-respiratory diseases as recently reviewed "PMID: 35114008 and 35206820 )
  • in the introduction authors should introduce the RAS pathway since it plays a key role in this manuscript 
  • "Increased ACE2 might be useful in the late stages of infection to counteract the hyperinflammatory and hyperphyrbotic state of lung tissue": How? 
  • "This may explain why SARS-CoV-2 infection was initiated by foods fed in the Wuhan market": authors must explain this association 
  • "Finally, increased ACE-2 could lead to increased bradykinin degradation, preventing pro-inflammatory, pro-oxidant, and profibrotic effects further.": Please, add reference 
  • abbreviations must be reported in full name when mentioned for the first time 
  • the role of aldosterone deserves to be discussed 
  • an accurate revision of english syntax and punctuation is necessary
  • References: authors must follow the journal style 

Author Response

Dear reviewer,
any comments have been accepted and changes made.
Thank you very much
The authors

Reviewer 2 Report

1.      RAS in Keywords. Abbreviations shall be avoided.

2.      The COVID-19 pandemic to date has caused about 120 Mln. Mln??

3.      Some reports regarding COVID need to be included in the Introduction section. PMID: 32441299; 32717346; 33577820.

4.      Increased expression of ACE2 could prove to be of therapeutic benefit in COVID-19 infection. Correct the sentence.

5.      Report is too short and shall be benefited in being a little extensive.

6.      Figures must be added for better and broader view.

Author Response

(The authors gave the same response as above.)

Round 2

Reviewer 1 Report

None of the concerns have been corrected. 

Changes must be clearly listed in the "author's reply file"  

Author Response

Major concerns

    17 out of 37 references of this manuscript are selfcitations. Many of them are not research articles but only point of view and letter to the editor. Authors must discuss original articles. ( reducing self-citations and adding more original articles, e.g. n 39,40,41 )    

 What is the aim of this study? Looking at the title, it seems that authors suggest possible therapies to target ACE2 in COVID-19 but this hypothesis is lost in the manuscript since they do not explain how these treatment can have a beneficial effect in COVID-19 patients. Moreover, authors conclude that "a pharmacological approach based on increasing ACE2 may reduce the detrimental actions mediated by stimulation of AT1r by Ang2, and increase the benefits of stimulation of MASr activated by Ang 1-7 and Ang 1-9". Authors must explain how increased ACE2 (it is not clear which form of ACE2 they mean) can induce this beneficial effects avoinding the side effects of these treatments. (The authors thank you for these interesting comments, we have made the changes in red and added Figure 2)

Paradoxically, use of ACEis and ARBs (commonly used in non-COVID-19 patients to treat hypertension) can increase ACE2 expression making these patients more susceptible to SARS-CoV-2 infection since the virus use ACE2 as receptor to enter into the cells. (changes in red and references n.42)

Minor concerns

    "2. The role of ACE-2": due to the fact that ACE2 is also expressed in many tissues, authors need to specify that SARS-CoV-2 can also cause non-respiratory diseases as recently reviewed "PMID: 35114008 and 35206820 ) (The authors consider  irrelevant to the purpose of the manuscript to add these citations.)

    in the introduction authors should introduce the RAS pathway since it plays a key role in this manuscript (inserted)

    "Increased ACE2 might be useful in the late stages of infection to counteract the hyperinflammatory and hyperphyrbotic state of lung tissue": How? (inserted)

    "This may explain why SARS-CoV-2 infection was initiated by foods fed in the Wuhan market": authors must explain this association (inserted)

    "Finally, increased ACE-2 could lead to increased bradykinin degradation, preventing pro-inflammatory, pro-oxidant, and profibrotic effects further.": Please, add reference (inserted)

    abbreviations must be reported in full name when mentioned for the first time (inserted)

    the role of aldosterone deserves to be discussed (not rilevant)

    an accurate revision of english syntax and punctuation is necessary (made)

    References: authors must follow the journal style(inserted)

Reviewer 2 Report

The authors are advised to address all the suggestions

Author Response

(The authors gave the same response as above.)

Round 3

Reviewer 1 Report

"2. The role of ACE-2": due to the fact that ACE2 is also expressed in many tissues, authors need to specify that SARS-CoV-2 can also cause non-respiratory diseases as recently reviewed "PMID: 35114008 and 35206820 ) (The authors consider irrelevant to the purpose of the manuscript to add these citations.)

It is relevant because authors totally ignored the fact that COVID-19 is not only a respiratory disease but can lead to many other complications 

References: authors must follow the journal style(inserted)

It looks like authors do not like the journal style since they still did not fix it

An accurate revision of syntax and typing errors is recommended

Author Response

"2. The role of ACE-2": due to the fact that ACE2 is also expressed in many tissues, authors need to specify that SARS-CoV-2 can also cause non-respiratory diseases as recently reviewed "PMID: 35114008 and 35206820 ) (The authors consider irrelevant to the purpose of the manuscript to add these citations.) References have been improved and added. The suggested reference is not relevant to this manuscript so it is not included.

It is relevant because authors totally ignored the fact that COVID-19 is not only a respiratory disease but can lead to many other complications. Accepted and added.

References: authors must follow the journal style(inserted) Accepted and amended.

It looks like authors do not like the journal style since they still did not fix it Accepted and amended.

An accurate revision of syntax and typing errors is recommended Accepted and amended.

Reviewer 2 Report

Some of the comments have still not been addressed. The manuscript can be accepted for publication after addressing all the comments.

Author Response

Some of the comments have still not been addressed. The manuscript can be accepted for publication after addressing all the comments.

Accepted. The whole text has been improved and modified.